# Molecular Cloning, In Silico Analysis, and Characterization of a Novel Cellulose Microfibril Swelling Gene Isolated from *Bacillus* sp. Strain AY8

**DOI:** 10.3390/microorganisms11122857

**Published:** 2023-11-24

**Authors:** Md. Azizul Haque, Dhirendra Nath Barman, Aminur Rahman, Md. Shohorab Hossain, Sibdas Ghosh, Most. Aynun Nahar, Mst. Nur-E-Nazmun Nahar, Joyanta K. Saha, Kye Man Cho, Han Dae Yun

**Affiliations:** 1Department of Biochemistry and Molecular Biology, Hajee Mohammad Danesh Science and Technology University, Dinajpur 5200, Bangladesh; shohorab1994@gmail.com (M.S.H.); aynun.jasmine@gmail.com (M.A.N.); nnnbhstu@gmail.com (M.N.-E.-N.N.); 2Department of Biotechnology and Genetic Engineering, Noakhali Science and Technology University, Noakhali 3814, Bangladesh; dhiren.bge@nstu.edu.bd; 3Department of Biomedical Sciences, College of Clinical Pharmacy, King Faisal University, Al-Ahsa 31982, Saudi Arabia; 4Department of Biochemistry and Molecular Biology, Trust University, Barisal 8200, Bangladesh; 5Department of Biological Sciences, College of Arts and Sciences, Carlow University, 3333 Fifth Avenue, Pittsburgh, PA 15213, USA; sghosh@carlow.edu; 6Department of Chemistry, Jagannath University, Dhaka 1100, Bangladesh; joys643@yahoo.com; 7Department of GreenBio Science and Agri-Food Bio Convergence Institute, Gyeongsang National University, Jinju 52725, Republic of Korea; 8Division of Applied Life Science (BK21 Program), Gyeongsang National University, Jinju 52725, Republic of Korea

**Keywords:** cellulose microfibril swelling gene, cloning and expression, H-bonds, in silico analysis, molecular cloning, recalcitrance, structural analysis

## Abstract

A novel cellulose microfibril swelling (Cms) gene of *Bacillus* sp. AY8 was successfully cloned and sequenced using a set of primers designed based on the conserved region of the gene from the genomic database. The molecular cloning of the Cms gene revealed that the gene consisted of 679 bp sequences encoding 225 amino acids. Further in silico analysis unveiled that the Cms gene contained the NlpC/P60 conserved region that exhibited a homology of 98% with the NlpC/P60 family proteins found in both the strains, *Burkholderialata* sp. and *Burkholderia vietnamiensis*. The recombinant Cms enzyme had a significant impact on the reduction of crystallinity indices (CrI) of various substrates including a 3%, a 3.97%, a 4.66%, and a substantial 14.07% for filter paper, defatted cotton fiber, avicel, and alpha cellulose, respectively. Additionally, notable changes in the spectral features were observed among the substrates treated with recombinant Cms enzymes compared to the untreated control. Specifically, there was a decrease in band intensities within the spectral regions of 3000–3450 cm^−1^, 2900 cm^−1^, 1429 cm^−1^, and 1371 cm^−1^ for the treated filter paper, cotton fiber, avicel, and alpha cellulose, respectively. Furthermore, the recombinant Cms enzyme exhibited a maximum cellulose swelling activity at a pH of 7.0 along with a temperature of 40 °C. The molecular docking data revealed that ligand molecules, such as cellobiose, dextrin, maltose 1-phosphate, and feruloyated xyloglucan, effectively bonded to the active site of the Cms enzyme. The molecular dynamics simulations of the Cms enzyme displayed stable interactions with cellobiose and dextrin molecules up to 100 ns. It is noteworthy to mention that the conserved region of the Cms enzyme did not match with those of the bioadditives like expansins and swollenin proteins. This study is the initial report of a bacterial cellulose microfibril swellase enzyme, which could potentially serve as an additive to enhance biofuel production by releasing fermentable sugars from cellulose.

## 1. Introduction

Cellulose saccharification, a critical step in cellulose degradation by cellulase, holds a key promise for bioethanol production. Bioadditives, expansin [1], and swollenin [2] proteins have demonstrated the ability to disrupt hydrogen bonds (H-bonds) in crystalline cellulose, thereby enhancing saccharification rates. In addition to bioadditives, scientists are actively searching for novel cellulose microfibril swelling (Cms) genes to have enhanced enzymatic saccharification activity to replace the chemical saccharification process. The Cms protein, a new bioadditive with an enzymatic activity, may offer a more efficient, cost-effective, and environmentally friendly approach for cellulose saccharification that can be scalable for industrial bioethanol production.

Cellulose, owing to its strong inter- and intra-molecular hydrogen bonding, we face with challenges in realizing its full potential in various applications, including fabricating desired forms or dissolving in inexpensive solvents [3]. Acid- and metal-base hydroxides have been known to partially swell cellulose by breaking H-bonds between adjacent molecules [4]. To this effect, the products of expansin genes, including α-expansin (E.X.P.A.), β-expansin (E.X.P.B.), Expansin-like A (E.X.L.A.), and Expansin-like B (E.X.L.B.) proteins, were identified in a variety of plant species [5] to have functions related to loosening the cell wall structure during plant cell growth. It has been proposed that these proteins disrupt non-covalent bonds between cellulose microfibrils and matrix polymers through a non-enzymatic mechanism, resulting in wall loosening and subsequent extension [6]. Furthermore, some expansins may play as a catalyst and potentially enhance the activities of hydrolytic enzymes responsible for degrading cell wall polymers during fruit ripening [5].

Swollenin, a protein capable of swelling cotton cellulose without producing detectable amounts of reducing sugars [3], has been found in bacteria [7] and ascomycete fungi [1,8,9]. The plant pathogen, *Clavibcater michiganensis* subsp. *sepedonicus* has been found to produce swollenin containing an endoglucanase domain and a domain with a sequence similar to that of expansins [1]. The recombinant swollenin from *Kluyveromyces lactis* has been reported to disrupt H-bonds and reduce cellulose crystallinity [2]. Collectively, these findings emphasize the potential roles of expansin and swollenin proteins in the bioengineering of cellulose via a chemical saccharification process, leading to bioethanol production. Another expansin-type protein called loosenin (LOOS1), identified in *Bjerkandera adusta*, was heterologously expressed, and the resulting products demonstrated the ability to bind polysaccharides [10]. LOOS1 facilitated the release of sugars from commercial cellulose and exhibited loosening activity on cotton cellulose [10]. Furthermore, the natural substrate, recalcitrant lignocellulosic material, when pretreated with LOOS1, became 7.5 times more susceptible to the action of a cocktail of enzymes, including cellulases and xylanases. Thus, LOOS1 has the potential to be used as a bioadditive enhancing enzymatic conversion processes [10] in the biotechnology industry. However, it should be noted that while the heterologous expression of expansins, swollenin, and loosenin proteins holds promise, knowledge about their effectiveness and application as bioadditives in bioethanol production appears to be limited and requires further explorations. Nonetheless, the discovery and characterization of these novel cellulose-modifying proteins provide scientists with new frontiers for advancing renewable energy research in the field of biotechnology. Further investigations optimizing the application of these proteins may ultimately lead to even more efficient and sustainable bioethanol production.

Cellulose-swelling proteins responsible for decreasing cellulose crystallinity and enhancing saccharification have been extensively studied in fungi, including *Trichoderma reesi* [1], *Trichoderma asperelumi* [11], *Bjerkandera adusta* [10], and Aspergillus fumigates [12]. However, only a few reports on bacterial proteins possess a similar ability to swell cellulose [13]. The commercialization of cellulose swelling using a chemical saccharification process mediated by expansin or swollenin has not been realized due to the high production cost of fungal proteins and the difficulties associated with optimizing their functional efficacies through bioengineering [2]. In addition, the physico-chemical properties along with the bioengineering potentials of cellulose-swelling proteins, including swollenin, expansin, and loosenin, have not been explored in-depth due to the lack of a sensitive analytical method. On the other hand, given that the bacterial cell growth rates are faster than those of fungi or plants, isolating a new bacterial strain capable of swelling cellulose microfibrils through H-bond disruption may be a significant advantage in the field of the biotechnology industry producing bioethanol.

This research aimed to characterize a cellulose microfibril swelling (Cms) gene isolated from *Bacillus* sp. strain AY8, collected from pulp sludge generated by a paper mill, that exhibited the ability to swell cellulose microfibrils via an enzymatic saccharification. A Congo red-based colorimetric method was developed and validated using the purified Cms enzyme [14] to measure the degree of cellulose swelling. The Cms enzyme demonstrated the disruption of cellulose intra-, inter-chain, and inter-sheet H-bonds, resulting in the swelling of cellulose microfibrils. Consequently, the enzyme was named cellulose microfibril swellase. This is the first report of an enzyme capable of swelling cellulose microfibrils, offering new insights into the bioengineering of cellulose with potential applications in environmentally desirable and scalable bioethanol production.

## 2. Materials and Methods

### 2.1. Materials

Filter paper (Whatman no. 1, Cat No 1001-150) and fat-free medical-grade cotton cellulose were procured from G.E. Healthcare U.K. Limited (Amersham Place, Buckinghamshire, UK) and TAE MYONG (Kyeung Buk, Daegu, Republic of Korea), respectively. Avicel, a cellulose powder derived from cotton linters, α-cellulose powder generated from wood, and Bradford reagents were purchased from Sigma-Aldrich Chemical (St. Louis, MO, USA) and Bio-Rad Lab, respectively. Ultrapure water, obtained through reverse osmosis of distilled water (3 D water milli Q filter) followed by ion exchange and filtration using the Milli-Q purification system from Millipore, was used to produce Tris-HCl buffer with a pH of 8.2. An H.P. 8453E UV-vis spectrophotometer was employed to determine light absorbance for the Bradford assays. The genomic DNA of the bacterial species was isolated from *Bacillus* sp. strain AY8 as described by Haque et al. (2015) [14] and subsequently stored at −80 °C until it was used in experiments.

### 2.2. List of Primers Used in PCR and DNA Sequencing

The primers used in this study are listed in Table 1.

### 2.3. Strains and Culture Conditions

Bacillus sp. strain AY8 was isolated from the pulp sludge generated by a paper mill (Moorim Paper, Jinju City, Republic of Korea), as previously described [14]. The composition of the filter paper swelling media was as follows: (NH_4_)_2_SO_4_, 0.1%; KH_2_PO_4_, 0.05%; K_2_HPO_4_, 0.05%; MgSO_4_.7H_2_O, 0.02%; CaCl_2_, 0.01%; NaCl, 0.01%; yeast extract, 0.1%; and filter paper, 0.2% [14,15]. The bacterial strain AY8 had secreted protease, DNase, chitinase, and cellulose microfibril swelling enzymes except cellulase [14]. The present study targeted to clone cellulose-swelling enzymes from the strain’s genome. To this end, DNA from the the strain AY8 was extracted, and the total genomic DNA was captured on a silica membrane in a spin column format, as described previously [16,17]. The DNA was washed, eluted, prepared for PCR analysis, and stored at −20 °C for future use [18].

### 2.4. Cms Gene Cloning

The protein database of *Bacillus cereus* Q1 was screened based on extracellular signal peptide and molecular mass around 16 to 17 kDa. The extracellular protein was confirmed using SignalP 4.1 Server (http://www.cbs.dtu.dk/services/SignalP/ (accessed on 10 February 2019)). Employing a similar strategy, the extracellular proteins from the protein database of the following bacterial strains were screened: *Bacillus thuringiensis serovar pulsiensis* B.G.S.C., *Ruminococcusobeum* ATCC 29174, *Clostridium bolteae* ATCC BAA-613, *Clostridium difficile* 630, *Mollicutes bacterium* D7, *Lachnospiraceae bacterium* 3_1, *Lactobacillus viri* DSM 20605, *Bacillus cereus* R309830, *Bacillus cereus* AH187, *Bacillus cereus* Rock1-15, *Bacillus cereus* BAG4X12-1, *Bacillus cereus* VD107, *Bacillus cereus* AH603, *Bacillus cereus* HuA2-1, *Bacillus cereus* VD142, and *Bacillus cereus* MSX-D12. To amplify the Cms homologs from the *Bacillus* sp. strain AY8, the PCR primer and the degenerate oligonucleotide primer were designed based on conserved amino acid sequences adjacent to the high conserved regions available in previously described databases [19]. The sense and antisense degenerate oligonucleotide primers were 5′-GAC TGC AGC GGC TTC GTG CGC TAC-3′ (sense) and 5′-GCC GAT RTA GAT GCC GAC GTG-3′ (antisense), respectively. The PCR amplification was conducted using *Bacillus* sp. strain AY8 DNA, super-Therm DNA polymerase (J.M.R., Side Cup, Kent, UK), 1.5 mM MgCl_2_, and 38 cycles of denaturing at 94 °C for 30 s, annealing at 64 °C for 30 s, and extension at 72 °C for 90 s. The anticipated product of approximately 200 bp was isolated from an agarose gel using a gel extraction kit (NucleoGen, Seoul, Republic of Korea). The chromosomal walking was conducted using the above-mentioned sense and antisense primers to obtain the sense and antisense primer of the complete open reading frame (ORF) of the Cms gene, as described previously [20].

### 2.5. Expression of Cms Gene in E. coli

For high expression of Cms gene, the PCR product generated with primer 2710F and 2711R was cloned into expression vector pET-26b(+) (Novegen, Dublin, Ireland) using *Bam*HI and *Hind*III sites, as described previously [21]. Briefly, *E. coli* BL21 (DE3) carrying pET-26b(+)/*Cms* was grown at 37 °C to mid-log phase in L.B. medium containing 50 µg/mL kanamycin. The cells were harvested via centrifugation (6000 g, 10 min), washed twice with and resuspended in 10 mMTris-HCl buffer (pH 7.0), and then stored at −20 °C until further use. The Bradford method [22] and SDS_PAGE [19] were used to determine the protein concentration and to separate protein molecules, respectively.

### 2.6. pH and Temperature Effects

The optical density (O.D.) of the dye, Congo red (C.R.) solution after adsorption by untreated cotton, or recombinant enzyme pretreated cotton were considered as control and test samples, respectively. The collected O.D. from the control to test sample was termed Congo red adsorption enhancement (C.A.E.). Thus, C.A.E. = O.D. of control solution—O.D. of test sample solution. The C.A.E. was converted to μM using the C.R. standard curve equation, *y =* 0.0103*x* − 0.0005 [14,23]. The effects of pH and temperatures on the recombinant Cms enzyme were analyzed, as described previously [14].

### 2.7. FT-IR and XRD Analyses of Cms-Treated Filter Paper

The effects of the recombinant Cms enzyme on cellulose unwinding were investigated using FT-IR [14,23] and XRD [24,25] analyses of the hydrogen bond disruption and the crystallinity changes, respectively, in various samples including filter paper, defatted cotton fiber, avicel, and alpha cellulose, as described previously. The FT-IR data in the spectral range from 800 to 4000 cm^−1^ were recorded for both untreated and Cms-treated samples to analyze the hydrogen bond disruption [14,23] using a powder FT-IR spectrometer (VERTEX 80v, Bruker Optics, Ettlingen, Germany). The XRD data for both untreated and treated and Cms-treated samples were collected using a powder X-ray diffractometer (GADDS, Bruker AXS, Karlsruhe, Germany) with Cu Kα radiation (λ = 0.154 nm), and the crystallinity changes were calculated using the following formula [24,25]:CrI = {(I_002_ − Iam) × 1_00_}/I_002_
where CrI is the crystallinity index, I_002_ is the maximum intensity of the I_002_ peak at 2θ = 22.5°, and I_am_ is the intensity at 2θ = 18.7°.

### 2.8. Molecular Docking and Visualization

PyRx software (http://pyrx.sourceforge.net/ (accessed on 10 December 2022)) was employed for conducting the molecular docking analysis, enabling the study and prediction of how ligands interact with macromolecules [16,19]. The Discovery Studio 2021 Client software (https://discover.3ds.com/discovery-studio-visualizer-download/ (accessed on 12 December 2022)) was utilized to create protein–ligand complexes and visualize their 2D and 3D structures using the chimera software. During the molecular docking process, the grid box size (X, Y, and Z) was fixed at 61.6411, 61.6072, and 61.8787, respectively, representing the maximum size of the box. The binding affinity of the protein with its ligands was computed using a negative score and expressed as kcal/mole [18].

### 2.9. Molecular Dynamic Simulations

The molecular dynamic simulations were performed with the GROMACS simulation package (https://www.gromacs.org/about.html/ (accessed on 12 December 2022) using the NPT ensemble under the temperature and the pressure set at 300 Kand 1 bar, respectively, for all runs as described previously [26]. The protein preparation wizard was used to prepare the protein–ligand complexes, followed by solvation using the orthorhombic simple point-charge water model through the system’s builder panel [27]. The resulting solvated system was then charged and neutralized by adding counter ions (Na^+^ or Cl^−^ ions) as required [27].

For the ligands cellobiose and Feruloyatedxyloglycan, the simulation length was set to 100 ns, with a relaxation time of 1 ps. As previously described [16,21], the molecular dynamic simulations were run using the NPT ensemble, an isothermal–isobaric ensemble with constant temperature, constant pressure, and a constant number of particles. The temperature and pressure were maintained at 300 K and 1.013 bars, respectively, utilizing the default settings for relaxation before the simulation [26,27].

## 3. Results

### 3.1. Gene Target and Cloning of the Cms Gene

The extracellular proteins, with a molecular weight of approximately 16 to 17 kDa, have been identified as members of the hypothetical NlpC/P60 protein family of the following bacterial strains: *Bacillus* sp.AY8, *Bacillus cereus* Q1, *Bacillus thuringiensis serovar pulsiensis* BGSC, *Bacillus cereus* R309830, *Bacillus cereus* AH187, *Bacillus cereus* Rock1-15, *Bacillus cereus* BAG4X12-1, *Bacillus cereus* VD107, *Bacillus cereus* AH603, *Bacillus cereus* HuA2-1, *Bacillus cereus* VD142, and *Bacillus cereus* MSX-D12. However, it was found that the NlpC/P60 family proteins from other strains, including *Ruminococcusobeum* ATCC 29174, *Clostridium bolteae* ATCC BAA-613, *Clostridium difficile* 630, *Mollicutes bacterium* D7, *Lachnospiraceae bacterium* 3_1, and *Lactobacillus viri* DSM 20605, have a different molecular mass, ranging from 33 kDa to 34 kDa (Table 2).

This study shows that a partial sequence of the conserved region of the NlpC/P60 gene of *Bacillus* sp. AY8 was selected for cloning. The cloning strategies, outlined in Figure 1, were employed, and the primers (2682F and 2683R) were designed based on the protein databases of *Bacillus cereus* Q1, *Bacillus cereus* NC7401, and *Bacillus cereus* AH187 (Figure 1). Additionally, a putative oxidoreductase (ORD) gene (1672 bp) was cloned and sequenced using primers (2672F and 2673R) based on the phylogenetic relationship (16S rDNA sequence) of *Bacillus* sp. AY8 with *Bacillus cereus* Q1, *Bacillus cereus* AH187, *Bacillus cereus* 03BB102, and *Bacillus cereus* AH820. The putative cloned gene sequence of *Bacillus* sp. AY8 showed 99% homology with *Bacillus cereus* Q1, *Bacillus cereus* NC7401, and *Bacillus cereus* AH187, respectively. This result helped us to deduce further to target the genomes of these three strains for cloning the NlpC/P60 gene from *Bacillus* sp. AY8.

The conserved region of the NlpC/P60 protein of Bacillus cereus Q1 exhibited similarities with the same proteins found in *Escherichia coli*, *Bacillus subtilis*, *Haemophilus influenzae*, *Lactococcus lactis*, *Salmonella typhimurium*, *Xylella fastidiosa*, and *Yersinia pestis*. Based on the protein homology, the primers 2682F and 2683R were designed and applied to *Bacillus* sp. AY8 genomic DNA for cloning the gene. However, the expected 192 bp PCR product band was not observed, even after using the temperature gradient PCR strategy (Figure 1). Importantly, the putative ORD gene PCR product band was observed from the same genomic DNA of *Bacillus* sp. AY8 (Figure 2). As a result, the focus shifted to hypothetical proteins NlpC/P60 of *Burkholderia* sp. 383, *Burkholderia cenocepacia* HI2424, and *Burkholderia cenocepacia* HI2424 due to their close phylogenetic relationships with *Bacillus* sp. AY8 (based on 16S rDNA). The NlpC/P60 proteins of these three *Burkholderia* strains showed higher homology with well-conserved [DCSGF] and [HVGIY] amino acid sequences (Figure 2). The conserved region of *Burkholderia* sp. proteins matched with other NlpC/P60 proteins of *Escherichia coli*, *Salmonella typhimurium*, *Yersinia pestis*, *Haemophilus influenzae*, *Xylella fastidiosa*, *Bacillus subtilis*, *Bacillus cereus* NC7401, and *Lactococcus lactis* (Figure 2).

Based on the conserved region, the primers 2684F and 2685R were designed, and the expected band size of the PCR product was 200 bp. These primers were applied to *Bacillus* sp. AY8 gDNA, resulting in the amplification of the gene with a PCR product band size of approximately 200 bp (Figure 2A). Subsequently, primers 2684F and 2685R were separately used for chromosome walking to amplify the gene, and finally, primers 2709F and 2708R were designed to obtain the complete ORF of the gene. Using this primer set, the expected PCR product band size of 0.678 kb was observed (Figure 2B).

### 3.2. Structure Analysis and Expression of the Cms Gene

The gene product was successfully cloned and sequenced, revealing a complete ORF with an ATG start codon and a TGA stop codon. The ORF, known as the Cms gene, spans 678 bp nucleotides and encodes a deduced protein comprising 225 amino acids (Figure 3). Notably, the deduced protein contains an extracellular signal peptide at its N-terminus, encompassing the first 32 amino acids (Figure 3).

Furthermore, the deduced protein exhibited a consensus region characteristic of the motor protein of the motA family, which was identified between amino acids 21 and 38 of the Cms enzyme. Potential ATP interacting residues were observed within the signal peptide sequence, motor protein consensus region, and conserved [DCSGF and HVGIY] region of the Cms enzyme. As determined using Compute pI/MW analysis (http://expasy.org/tools/pi_tool.html (accessed on 20 December 2022)), the predicted Cms protein (225 aa) has a theoretical estimated molecular mass of 24,248.11 Da, while the signal peptide alone has a molecular mass of 3291.84 Da. Interestingly, when subjected to SDS-PAGE, the molecular mass of the enzyme was estimated to be approximately 19.5 kDa (Figure 4) compared to that of a putative mass of 24 kDa.

### 3.3. Comparision of Amino Acids of Cms Enzyme of Bacillus sp. AY8 with Those of Other NlpC/P60 Family Proteins

The amino acid sequence of the Cms enzyme from *Bacillus* sp. AY8 was compared with those found in other bacterial strains (Figure 5). To identify closely related Cms proteins, we utilized standard search algorithms (specifically, FASTA and BLAST programs) to search protein sequence databases, including SWISS-PORT, PIR, and GenBank. Subsequently, we employed DNAMAN analysis systems to construct a phylogenetic tree of the NlpC/P60 proteins from different bacterial strains, as presented in Figure 5. The resulting phylogenetic tree indicated that the Cms enzyme shares a close relationship with the NlpC/P60 proteins of *Burkholderia* sp. However, it displayed significantly low similarity compared to the NlpC/P60 proteins of *Bacillus cereus*. This indicated that the Cms enzyme is more closely related to *Burkholderia* sp. than to *Bacillus cereus* within the NlpC/P60 protein family.

The multiple sequence alignment analyses revealed that the Cms enzyme shares 23.4% homology with expansin, swollenin, and loosenin proteins (Figure 5). Notably, the Cms enzyme contains a conserved region characteristic of NlpC/P60 family proteins. However, this conserved region is absent in other cellulose-disrupting proteins like swollenin, expansin, and loosenin. These findings strongly suggest that the Cms enzyme belongs to the NlpC/P60 family and possesses cellulose microfibril swelling activity. The presence of the conserved region in Cms, which is absent in other cellulose-disrupting proteins, underscores its putative distinct role and function related to cellulose degradation and/or modification. The observed homology among Cms, expansin, swollenin, and loosenin proteins further suggests some standard features or evolutionary threads among these enzymes, despite their differences in the conserved NlpC/P60 family region.

#### 3.3.1. Molecular Docking Analysis of Cms Enzyme

Three small ligand molecules, namely cellobiose, dextrin, and feruloyatedxyloglucan, were docked with the Cms enzyme. The docking scores for the molecules ranged from −5.8 to −6.2 kcal/mol, indicating favorable binding affinity between the small molecules and the Cms enzyme.

Cms–cellobiose docking analysis revealed the multiple molecular interactions between cellulose and the Cms protein with several amino acid residues (Figure 6A). The O-atoms in the phosphodiester bond of cellulose formed conventional hydrogen bonds with Asp164, Tyr161, and Asn165. H-atoms were attacked by Ser138, His3, and Leu140 through conventional hydrogen bonds and carbon–hydrogen bonds. Interestingly, the residues Ser138, His3, and Asp164 in the binding pocket of the Cms enzyme formed a possible catalytic triad. The catalytic site residue interaction distances were measured to be less than 3 Å, explaining the strong binding affinity between cellulose and the Cms enzyme.

Cms–dextrin docking analysis showed significant interactions between the Cms enzyme and dextrin compound involving several amino acid residues (Figure 6B). Ser138, Tyr161, Asn139, and Leu140 interacted with the O and H atoms of the dextrin molecule through conventional hydrogen bonds and carbon–hydrogen bonds. Arg2, Ala71, Asp183, and His3 residues were also connected with the O and H atoms of Dextrin. Notably, His3, Asp183, and Ser138 constituted the catalytic triad in the Cms–dextrin complex, with interaction distances within <5 Å.

Cms–Feruloyatedxyloglucan docking analysis revealed three types of interactions between Cms and Feruloyatedxyloglucan: (i) conventional hydrogen bonds, (ii) carbon–hydrogen bonds, and (iii) Pi–alkyl bonds mediated by Leu140, Tyr161, Asp164, Asn165, Asp183, Gly163, and Phe69 amino acid residues. The interaction distance between protein–ligand molecules was observed within <5 Å (Figure 6C).

The binding affinities of all three test ligand molecules with the Cms enzyme were found to be −5.8, −6.1, and −6.2, respectively. The interacting amino acids of the docking complexes for these molecules were remarkably similar to the second site of the protein cavity predicted by the Ftsite server. Molecular docking analysis provided valuable information about the interactions between the Cms enzyme and various small molecules (putative ligands). The analyses also suggested that the test molecules can effectively bind to the active site of the Cms enzyme.

#### 3.3.2. Molecular Dynamics Simulations

The stability of the Cms–ligand complexes was assessed by comparing the root mean square deviation (RMSD), root mean square fluctuation (RMSF), and radius of gyration (Rg) values with those of the unbound protein structure during molecular dynamics simulations.

The RMSD plot of the Cms–dextrin complex initially showed an increasing trend from 0 to 7 ns, with an RMSD value ranging from 0.4 to 1.35 nm (Figure 7A). After 8 ns, the RMSD remained stable, with an average value of 1.4–1.6 nm up to 100 ns. In contrast, the RMSD value of the Cms–cellobiose complex fluctuated between 0.8 and 1.5 nm throughout the 100 ns simulation. The difference in RMSD values between Cms and the Cms–ligand complexes was approximately 0.8–1.32 nm for the cellobiose complex and 0.4–0.7 nm for the dextrin complex (Figure 7A).

The RMSF plot of the Cms–cellobiose complex showed favorable residual interactions between the Cms protein and cellobiose atoms at specific amino acid residues, such as Ser106, Val109, Val112, Phe113, Thr116, Leu117, Gly118, and Met119, along with residues Ser138, Leu140, Tyr161, Asp164, and Asn165 in close proximity. In the Cms–dextrin complex, residues Phen108, Val109, Gly143, His157, Val158, Gly159, Ile160, Val168, His169, and Ser170 showed low RMSF values close to the baseline (~0.11–0.15 nm), except Cys105 and Ser106 (<0.15 nm). Favorable residual interactions between the Cms protein and dextrin atoms were identified at Ser106, Val109, Val112, Phe113, Thr116, Leu117, Gly118, and Met119 (Figure 7B).

The Rg values for only Cms, Cms and cellobiose, and Cms and Dextrin fluctuated between 1.8 and 2.35 nm until 25 ns, with the Cms–dextrin complex achieving stability and compactness at 1.8 nm after 25 ns. For Cms and cellobiose, Rg values showed fluctuations between 2.08 and 2.42 nm from 30 to 70 ns, followed by further fluctuations between 2.4 and 2.7 nm up to 90 ns and remaining stable until 100 ns. The Rg values for Cms alone exhibited fluctuations between 2.1 and 2.6 nm until 703 ns, followed by fluctuations between 2.2 and 2.45 nm up to 100 ns (Figure 7C).

The fluctuation in Rg values can be attributed to the flexibility of the protein–ligand complex during the molecular dynamic simulation. However, all three systems (only the protein and the protein–ligand complexes) exhibited relatively consistent and similar values for Rg, confirming their stable conformations. Overall, the RMSD, RMSF, and Rg analyses suggested that the Cms–ligand complexes undergo stable dynamics during the molecular dynamic simulations, supporting the reliability and robustness of the binding interactions between the Cms enzyme and its putative ligands (cellobiose and Dextrin).

### 3.4. Cellulose Swelling Evidence of Recombinant Cms Enzyme on Cellulose Substrates

The FT-IR spectrum analysis of filter paper cellulose (Figure 8A) revealed a noticeable decrease in band intensity within the 3000–4000 cm^−1^ range when treated with the Cms enzyme compared to that of untreated (control). The observed decline can be attributed to the weakening of both intra- and inter-chain H-bonds, leading to a tightening effect on the OH vibrations. Consequently, the vibrational amplitudes and oscillator strengths were reduced. The spectral region from 3000 to 4000 cm^−1^ is particularly valuable for understanding the H-bonding patterns of cellulose, and the decrease in peak intensity following Cms enzyme treatment of the filter paper indicated the disruption of H-bonds. Additionally, the bands at 2900 cm^−1^, 1429 cm^−1^, and 1371 cm^−1^, known for their sensitivity to the crystallinity state of cellulose, showed reduced intensity following Cms enzyme treatment. This suggests an increase in the proportion of amorphous cellulose in the filter paper cellulose. Similarly, the peak intensities corresponding to the H-bonds were declined in the spectra of the Cms enzyme-treated samples of defatted cotton fiber (Figure 8B), avicel (Figure 8C), and alpha cellulose (Figure 8C), indicating the disruption of H-bonds in cellulose.

The Cms enzyme treatments displayed different diffraction patterns and crystallinity index (CrI) based on the sources of cellulose materials. The intensity of the 002 plane was found to be lower in the Cms-treated cellulose of filter paper compared to that of the untreated filter paper. The CrI values for the untreated and Cms-treated filter paper were determined to be 95.51% and 92.6%, respectively (Figure 9A). In defatted cotton fibers, Cms treatment caused a reduction in the intensity of the 002 plane, leading to a decrease in the CrI value from 85.01% in untreated to 81.04% (Figure 9B). In contrast, the intensities of the 002 planes in both Cms-treated commercial grades of avicel and alpha cellulose significantly declined compared to those in untreated samples. Consequently, the CrI values of the treated avicel (Figure 9C) and α- cellulose (Figure 9D) were reduced from 93.28% (untreated control) to 88.62% and 98.91% (untreated control) to 84.94%, respectively. These findings suggest that Cms enzyme treatment may disrupt the H-bonds between adjacent cellulose sheets, potentially leading to the separation of parallel sheets in our test samples including filter paper microfibrils, cotton fibers, avicel, and –α- cellulose.

## 4. Discussion

H-bonds significantly hinder the enzymatic hydrolysis of cellulose-releasing fermentable sugars for biofuel production. Enhancing the accessibility of hydrolytic enzymes to cellulose by loosening or swelling the cellulose surface through amorphogenesis is a promising strategy [10,12,14]. One such molecule, swollenin, an expansin-like protein first identified in *Trichoderma reesei* (a mesophilic and filamentous fungus), has shown unique characteristics, including disruption of Valonia (singe-celled marine algae) cell wall fragments and weakening of filter paper, but without the detectable release of reducing sugars [11]. Cellulose-swelling proteins have been mainly isolated from fungi or plants, making their large-scale production cumbersome and commercially unscalable. To this effect, cellulose-degrading enzymes are of immense interest in biotechnological applications due to their potential to facilitate biomass conversion leading to biofuel production. Thus, the present study focused on characterizing the Cms gene from the bacteria, *Bacillus* sp. AY8. Understanding the molecular and functional properties of Cms could yield valuable insights into its cellulose-swelling activity and potential industrial applications for scalable bioethanol production.

In this study, we have cloned the gene from *Bacillus* sp. AY8, encoding the Cms enzyme. The gene has been identified as one of the members of the hypothetical protein, NlpC/P60 family, in *Bacillus* sp. AY8 and related strains. Interestingly, while the molecular weight of these extracellular proteins from *Bacillus* sp. AY8 and related strains ranged from 16 to 17 kDa, it is much smaller than those from other bacteria (*Ruminococcusobeum* ATCC 29174 and *Clostridium difficile* 630) with similar expansin-like protein activity, with a molecular mass of approximately33 to 34 kDa. This difference in molecular weight raised questions about potential functional variations among these enzymes and warranted the targeted cloning of the NlpC/P60 gene from *Bacillus* sp. AY8. Strategically, the conserved region of the NlpC/P60 gene from *Bacillus* sp. AY8 was selected and cloned, as discussed above. The cloned gene sequence of *Bacillus* sp. AY8 showed a high degree of homology with *Bacillus cereus* Q1, *Bacillus cereus* NC7401, and *Bacillus cereus* AH187, providing further support for the selection of these strains for subsequent cloning of the NlpC/P60 gene.

Furthermore, a comparative analysis of the amino acid sequence of the Cms enzyme with other NlpC/P60 family proteins revealed its close phylogenetic relationship with the proteins of *Burkholderia* sp. rather than those of *Bacillus cereus.* This observation suggests that the Cms enzyme may share functional similarities with the NlpC/P60 proteins of Burkholderia sp., possibly involved in cellulose degradation and/or modification. The alignment of the conserved region of the NlpC/P60 protein from *Bacillus cereus* Q1 revealed similarities with those proteins from other bacterial species like *Escherichia coli, Salmonella typhimurium*, and *Bacillus subtilis*, among others. This suggests potential functional and evolutionary relationships among these cellulose-degrading enzymes. The comparison of the Cms enzyme amino acid sequence with other cellulose-disrupting proteins, such as expansin, swollenin, and loosenin revealed a 23.4% homology, indicating some standard features or evolutionary relationships among these enzymes. However, the presence of a conserved region unique to NlpC/P60 family proteins in the Cms enzyme underscores its distinct role and function in cellulose degradation and/or modification [28].

Structural analysis of the Cms enzyme revealed the presence of a consensus region characteristic of the motor protein motA family between amino acids 21 and 38, potentially contributing to the enzyme’s cellulose-swelling activity. The predicted molecular mass of the Cms enzyme was approximately 24.2 kDa, which was inconsistent with the estimated mass obtained from SDS-PAGE analysis (Figure 4A). The molecular mass of the putative Cms gene product was estimated to be around 19.5 kDa, which is approximately 2.5 kDa higher than the purified Cms enzyme, potentially due to proteolytic cleavage loosing the signal peptide during production. The molecular mass of the normal enzyme from several *Bacillus* sp. was lower than the predicted molecular mass of the cloned gene products—for example, the molecular mass of the normal proteinase enzyme from *Bacillus* sp. AK.1 was estimated to be 36.9 kDa, but its inactive proenzyme molecular mass, 45 kDa, was observed when expressed in *E. coli*. Later, the inactive proenzyme was activated in vitro after heating at 75 °C by removing the accessory peptide sequence [29]. The *Bacillus subtilis* gene product was determined to have a molecular mass of 60 kDa, but the expression of this gene resulted in the production of a 37 kDa enzyme in the culture medium, suggesting that the enzyme is synthesized in a preproenzyme form [30].

Moreover, the natural, mature protein has a predicted molecular mass of 68.197 kDa, but the isolated protein has an apparent molecular mass of 28.5 kDa due to C-terminal processing or proteolysis [31]. The conversion of the primary gene product into the mature enzyme was mediated by active subtilisin proteases; therefore, this processing is most likely autocatalytic [32]. The *E. coli* host cellular environment is different than *Bacillus* sp. AY8; therefore, it is supposed that the Cms gene product was probably partially processed in the host, *E. coli*. Consequently, a higher molecular mass of Cms was observed along with lower activity than the normal enzyme. However, experimental evidence is limited in this study to confirm whether the Cms enzyme contains any proenzyme sequence or undergoes any proteolysis.

Similar to expansin and swollenin, the filter paper was shown to be weakened by the Cms enzyme activity. The filter paper, avicel, and alpha cellulose consists of relatively pure cellulose. Therefore, weakening filter paper, avicel, and alpha cellulose could open the microfibers of cellulose by breaking the H-bonds [33]. No reducing sugar was liberated from the Cms-treated filter paper, which was in accordance with plant expansin activity [14]. Additionally, it is worth noting that the swelling of defatted cotton fiber did not result in the release of any reducing sugar. However, it did enhance the adsorption of Congo red, indicating the disruption of hydrogen bonds within cellulose microfibrils [14,23]. These findings align with a previous report on cellulose microfibril swelling enzymes by Haque et al. [14]. Nucleic acid helicase is an enzyme that translocates directionally through double stranded nucleic acid substrates to catalyze the separation (unwinding) of complementary strands [34,35]. Helicase is functionally a DNA motor protein that utilizes ATP energy [35]. However, motor protein enzymes convert the chemical free energy obtained from the hydrolysis of ATP into mechanical work to drive the process [36]. Consequently, these motor proteins provide useful models for the directional ATP-driven helicase translocation along ssDNA lattices. Analysis of *Bradhyrizobium* sp. S23321 flagellar motor protein revealed that the motA family consensus sequence, {A-[LMF]-x-[GAT]-T-[LIVMF]-x-G-x-[LIVMF]-x(7)-P}, was partly conserved in the N-terminal sequence (first 21 to 38 amino acids) of the Cms enzyme. In addition, the amino acid sequences of Cms contains potential ATP-interacting residues. Thus, it is possible that Cms has motor enzyme function that might separate cellulose microfibrils like helicase or motor enzymes action to DNA. However, the Cms enzyme has an NlpC/P60 family protein conserved region and less homology with plant expansins or swollenin, which could break H-bonds rather than β-(1–4) glycosidic linkages of cellulose [37].

The treatments involving Cms enzymes had a notable impact on the cellulose structure as manifested by weakened both intra (2-OH• • •O-6 and 3-OH• • •O-5) and intermolecular H-bonds (6-OH• • •O-3′), resulting in the tightening of OH vibrations. The reduction in vibrational amplitudes led to a decrease in oscillator strengths, as observed through a reduction in the peak intensity of cellulose within this region [13]. Furthermore, the impact of the Cms enzyme on cellulose structure was reminiscent of changes observed when cellulose underwent expansion due to an increase in temperature [36]. This expansion is associated with the breaking of H-bonds, as reflected through high intensity peaks in Cms enzyme-treated samples. Similar shifts in the region around 3300–3400 cm^−1^ were reported in the literature for the swelling of bleached pulp cellulose [38]. In our study, the untreated cellulose peak at 3409 cm^−1^ gradually shifted following Cms treatment to a higher wave number of 3425 cm^−1^ (Figure 8C). The observed shift in peak position, along with the decreased intensities of the H-bonded OH group vibrations in samples treated with the Cms enzyme can be attributed to the disrupted/broken H-bonds in cellulose, underscoring the significant impact of the Cms enzyme in creating an access for cellulase. The -OH stretching region within the spectrum, spanning from 3000 to 4000 cm^−1^, serves as a valuable tool for elucidating cellulose’s hydrogen bonding patterns, as highlighted in prior research [39]. Upon treatment with cellulose-swelling proteins [22], we observed a reduction in the peak intensity within this region, indicative of a disruption in hydrogen bonds. Furthermore, specific bands at 2900 cm^−1^, 1429 cm^−1^, and 1371 cm^−1^ are known for their sensitivity to crystallinity states [39]. Consequently, the diminished intensity of these peaks following Cms treatment suggests an increased presence of amorphous cellulose within the cotton structure.

XRD analysis revealed that the 002 planes corresponded to layered sheets of parallel chains, representing the most intense signal among cellulose polymorphs [15,16]. Thus, reduced intensities of 200 planes, in Cms-treated test samples resulted from the reduction of the strength of inter-sheet H-bonds by disrupting the H-bonds, O-6-H• • •O3′ along with a decrease in CRI, causing cellulose swelling that can be attributed to disruption of H-bonds as previously observed in the presence of swollenin and non-hydrolytic proteins [14]. Within the XRD spectra, the 002 plane stands out as it comprises layered sheets of parallel chains, boasting the highest intensity among various cellulose polymorphs [30]. This observation leads to a plausible inference: Cms treatment may induce the dissociation of the parallel sheets within cotton microfibrils by disrupting the inter-sheet hydrogen bonds. Consequently, a reduction in the CrI (crystallinity index) of cellulose becomes apparent, attributed to the disruption of hydrogen bonds facilitated by swollenin and non-hydrolytic proteins [2,13]. FT-IR, XRD, and solid-state NMR are sophisticated techniques for assessing cellulose structure, enabling precise measurements of hydrogen bonding and crystallinity, as referenced in prior studies [13,22,40,41,42]. Hence, the analysis of cellulose substrates treated with recombinant Cms using FT-IR and XRD can offer insights into the disruptive effects on hydrogen bonds. As reported previously [14,23], visual observations (Congo red analysis) of test samples treated with recombinant Cms enzymes showed a rough, unwound, and amorphous surface along with the expansion of cellulose swelling due to the dispersion of microfibrils. The effects of temperatures and pH on Cms enzyme activities were found to be modulated and differed from those reported for expansins, swollenins, and loosenins [1,10,12]. The differential enzyme activities can be attributed to the reaction milieu, modulated by temperatures and/or pH, which may play a key role in altering structural conformation and facilitating cellulose microfibril swelling. To understand the effects of the Cms enzyme on cellulose microstructure, filter paper cellulose was chosen for instrumental analysis due to its higher polymerization and abundant H-bonds. It is important to emphasize that the ability of the Cms enzyme to weaken filter paper, defatted cotton fiber, avicel, and alpha cellulose without releasing reducing sugars aligns with the behavior of plant expansins [38]. Interestingly, the presence of potential ATP-interacting residues in the amino acid sequences of the Cms enzyme and the conservation of its certain motifs related to motor proteins like helicases suggest that the enzyme might function as a motor enzyme that separates cellulose microfibrils. The unique characteristics of Cms, distinct from plant expansins or swollenins, underscores its potential role in disrupting H-bonds rather than β-(1-4) glycosidic linkages in cellulose [11].

This study also conducted molecular docking experiments to explore the interactions between the Cms enzyme and various ligands comprised of small molecules such as, cellulose, Dextrin, maltose 1-phosphate, and feruloyatedxyloglucan. The molecular docking analyses revealed favorable binding affinities between the Cms enzyme and test ligands, suggesting its ability to interact with diverse substrates [43]. The presence of a potential catalytic triad within the binding pocket of the Cms enzyme further supports its enzymatic activity in cellulose swelling. The molecular dynamics of the Cms–ligand complexes remained stable and compact throughout the simulation, indicating the robustness of the enzyme–substrate interactions, validating the reliability of the docking results [16] and providing a basis for potential optimization of application in biotechnology industry [21]. The successful molecular cloning of the Cms gene and the in silico structural characterization of the Cms enzyme using molecular docking analysis and MD simulation provided valuable insights into the enzymatic cellulose-swelling activity and its potential biotechnological applications. The Cms enzyme is unique in its ability to efficiently unwind the crystalline structures of cellulose microfibrils. This distinct mode of action enhances the accessibility of cellulase to cellulose microfibril, leading to reduced energy input and improved enzymatic hydrolysis of cellulose. In addition, the Cms enzyme exhibits synergistic effects with other enzymes, contributing to a speedy degradation of cellulose. The action versatility of the Cms enzyme makes it an ideal candidate for various processes requiring cellulose swelling and subsequent degradation. The advantages of the Cms enzyme over conventional cellulolytic enzymes include significantly increased efficacy of cellulolytic processes with reduced dosage of the enzyme along with a broader substrate range, encompassing pretreated biomass and recalcitrant feedstocks. Thus, the Cms enzyme might have a potential use in sustainable productions of biofuels from non-food biomass sources, textiles, and papers.

This study focused on the interactions between the Cms enzyme and a few potential ligand molecules (cellulose, Dextrin, and feruloyatedxyloglucan). Thus, further investigations with a broader range of cellulose derivatives and related compounds could provide a more comprehensive understanding of the enzyme’s substrate specificity. The functional studies, such as enzymatic assays and kinetic analysis, are yet to be conducted, as these would provide a direct measure of cellulose-swelling activity catalyzed by the Cms enzyme. The functional studies would strengthen the evidence for the enzyme’s biological activity and cellulose-modifying capabilities.

## 5. Conclusions

This study is the first to report a naturally occurring bacterial cellulose microfibril swellase (Cms) enzyme that can alleviate cellulose recalcitrance by disrupting H-bonds. Also, this study successfully cloned and characterized the Cms gene from *Bacillus* sp. AY8. In silico analyses of the Cms gene, including molecular docking and molecular dynamic simulations of the Cms protein, demonstrated the structure featuring amino acid sequences and potential sites for substrate interactions. Furthermore, the Cms enzyme exhibited cellulose-swelling activity, opening valuable insights into its potential use in biofuel production and laying the foundation for future investigations into its industrial applications. This study contributes to the field of biotechnology and holds promise for the development of eco-friendly approaches in the field of cellulose-based biofuel. However, the study has shortfalls, including limited substrate specificity exploration, lack of in vivo validation, and a need for functional assays to verify enzymatic activity. Our findings pave the way for further research to optimize the catalytic activities of the Cms enzyme and substrate specificity and potential applications in biomass conversion into biofuel production, among other sustainable processes.

## Figures and Tables

**Figure 1 microorganisms-11-02857-f001:**
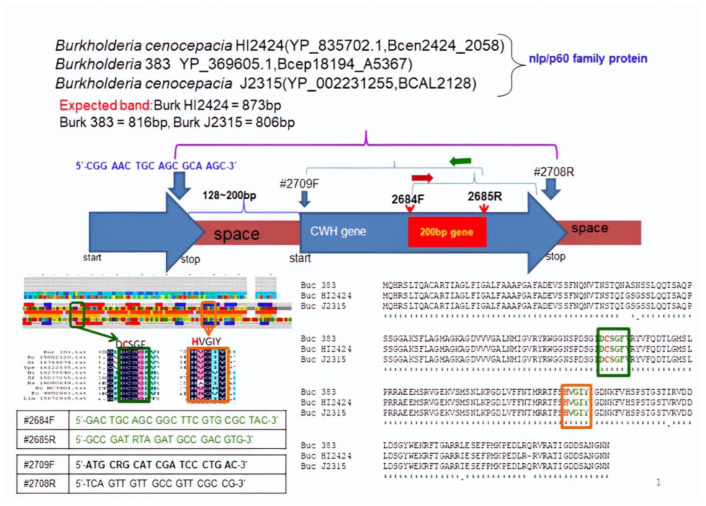
Cloning strategies of Cms gene from *Bacillus* sp. strain AY8.

**Figure 2 microorganisms-11-02857-f002:**
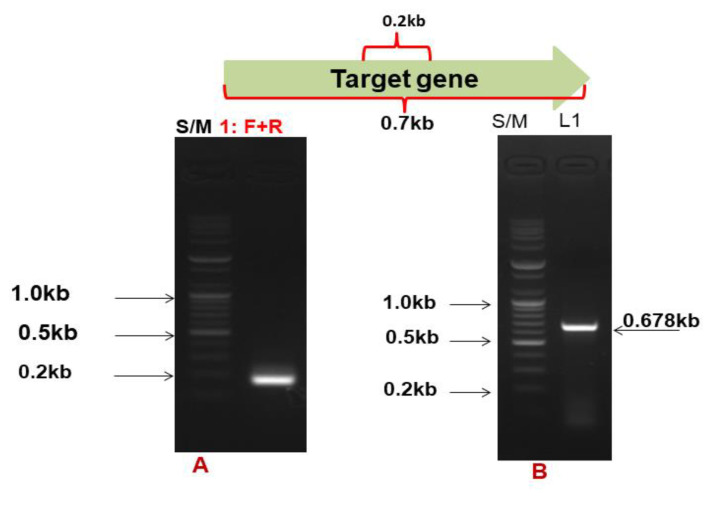
Agarose gel (1%) electrophoresis of (**A**) #2684F, #2685R products using. S/M, size mark; Lane 1, #2684F + #2685R (0.2 kb); (**B**) #2709F + #2708R products. S/M, size mark; Lane 1, #2709F + #2708R (0.678 kb).

**Figure 3 microorganisms-11-02857-f003:**
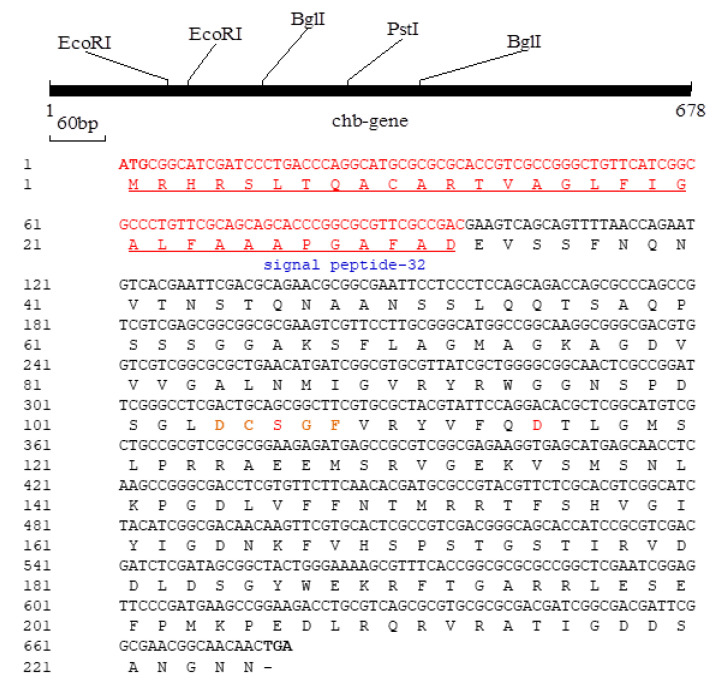
Nucleotide and deduced amino acid sequences of Cms gene of *Bacillus* sp. AY8. The stop codon is indicated by a bar, while the signal peptide is depicted by underline.

**Figure 4 microorganisms-11-02857-f004:**
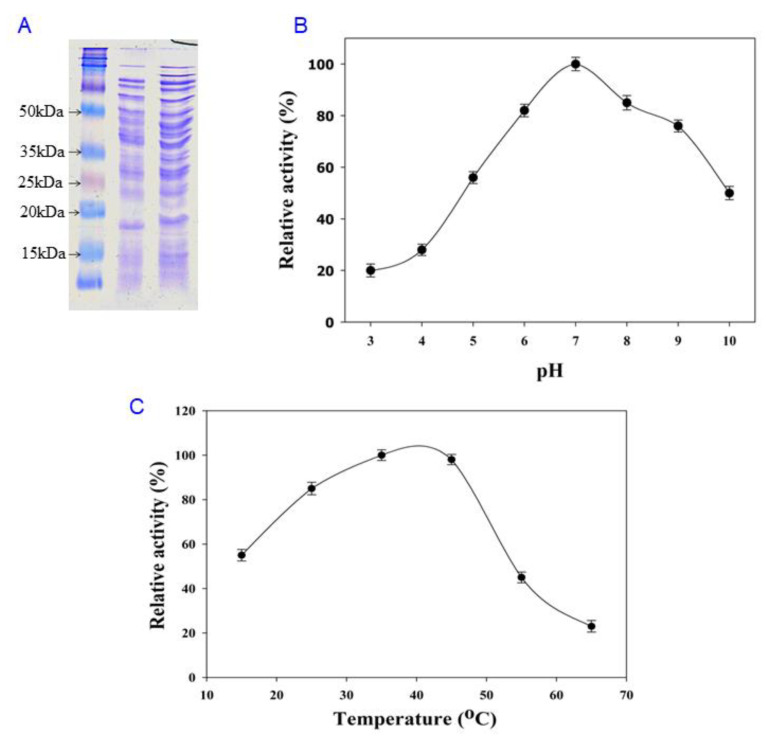
Expression of Cms gene of *Bacillus* sp. strain AY8. (**A**) Products Cms gene expression on SDS-PAGE; (**B**) pH; and (**C**) heat stabilities of Cms protein on Congo red adsorption enhancement on cotton cellulose, respectively [14].

**Figure 5 microorganisms-11-02857-f005:**
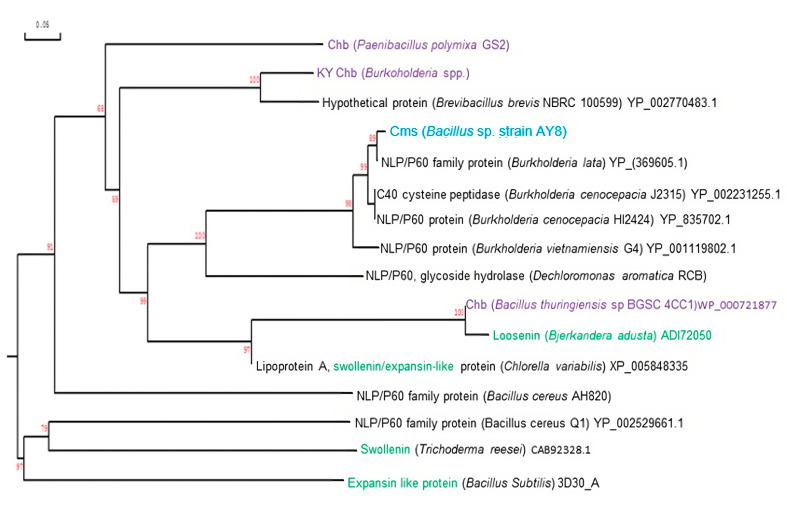
Phylogenetic relationships of *Bacillus* sp. strain AY8 Cms enzyme (blue) and other closely related proteins. Proteins highlighted with green color are cellulose swelling proteins involved in cellulose H-bonds disruption, whereas purple color are very similar proteins (unpublished) like Cms.

**Figure 6 microorganisms-11-02857-f006:**
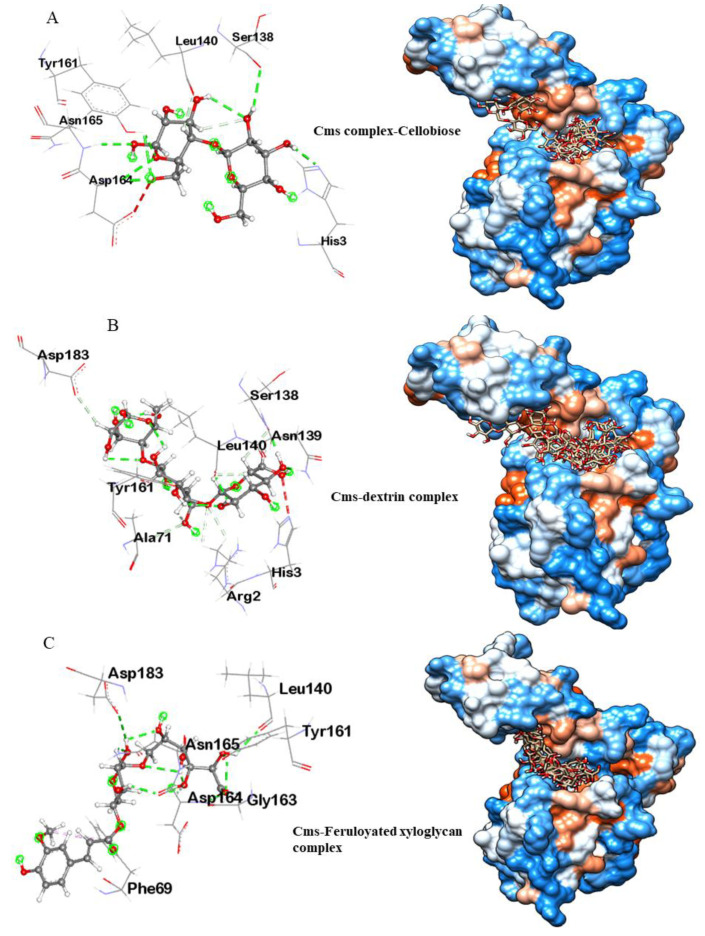
(**A**). Structure of Cms–cellobiose docking complex. (**B**). Structure of Cms–dextrin docking complex. (**C**). Structure of Cms–feruloyatedxyloglycan complex.

**Figure 7 microorganisms-11-02857-f007:**
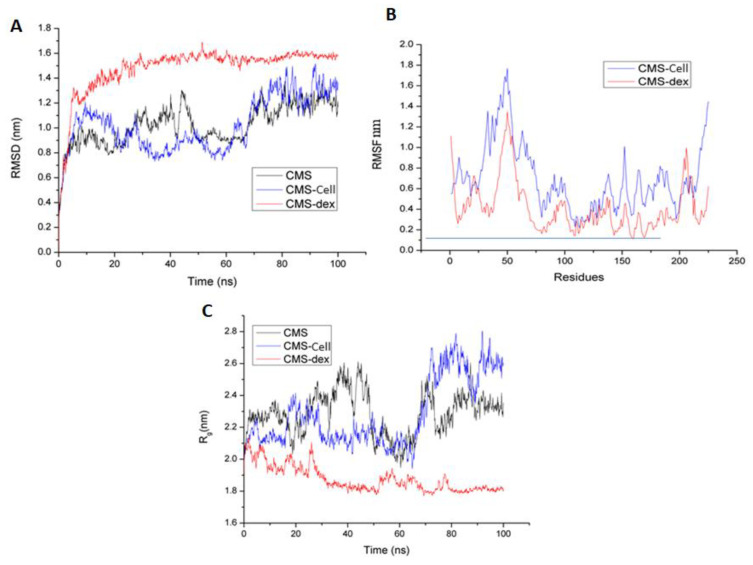
Molecular dynamic simulations of Cms proteins with cellobiose and dextrin complexes (**A**) Root mean square deviation (RMSD) of Cms–cellobiose and Cms–dextrin complexes, (**B**) root mean square fluctuation (RMSD) of Cms–cellobiose and Cms–dextrin complexes, (**C**) radius of Gyration (Rg) of Cms–cellobioase and Cms–dextrin complexes.

**Figure 8 microorganisms-11-02857-f008:**
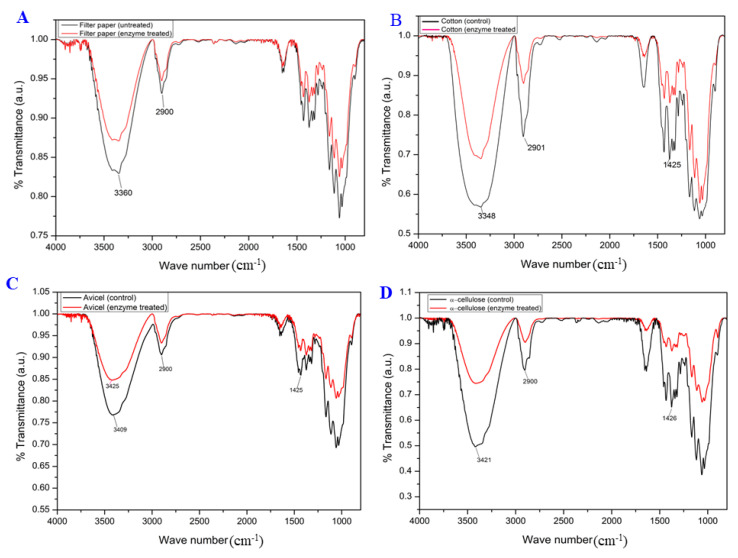
FTIR spectra of untreated and recombinant Cms enzyme-treated (**A**) filter paper, (**B**) cotton fiber, (**C**) avicel, (**D**) alpha cellulose. Samples of untreated or treated with recombinant Cms enzyme in Tris-HCl buffer (pH 7) were incubated at 37 °C in water bath for 24 h.

**Figure 9 microorganisms-11-02857-f009:**
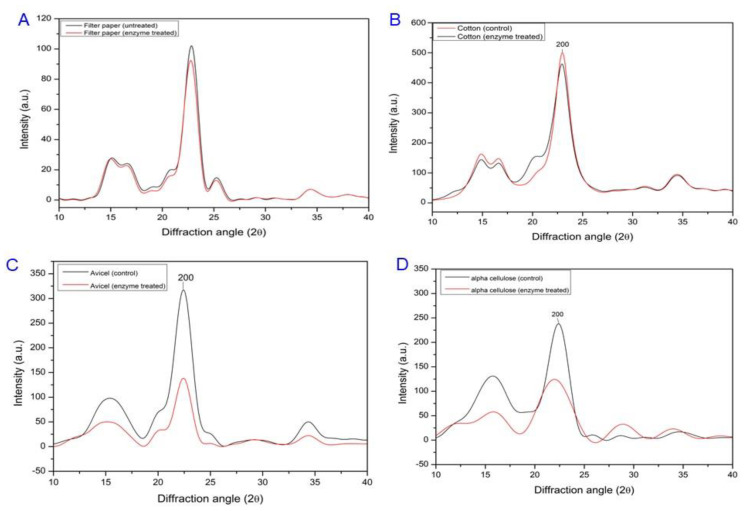
XRD spectra of untreated and recombinant Cms enzyme-treated (**A**) filter paper, (**B**) cotton fiber, (**C**) avicel, (**D**) alpha cellulose. Samples of untreated or treated with recombinant Cms enzyme in Tris-HCl buffer (pH 7) were incubated at 37 °C in water bath for 24 h.

**Table 1 microorganisms-11-02857-t001:** List of Primers.

Number	Uses	Sequence (5′→3′)
#2672F	Cloning	CCC AAT ACA ATT GCC AGC AG
#2673R	Cloning	GCT ATT TCC AAC ATT CCG RAG C
#2682F	Cloning	GGR TTT GAT TGT TCG GGR TTA
#2683R	Cloning	GCC AAT ATA CAT ACC AAC GTG
#2684F	Cloning	GAC TGC AGC GGC TTC GTG CGC TAC
#2685R	Cloning	GCC GAT RTA GAT GCC GAC GTG
#2709F	Cloning	ATG CRG CAT CGA TCC CTG AC
#2708R	Cloning	TCA GTT GTT GCC GTT CGC CG
#2710F	Cloning	GGA TCCGAAGTCAGCAGTTTTA
#2711R	Cloning	AAG CTT GTTGTTGCCGTTCGC

**Table 2 microorganisms-11-02857-t002:** List of strains used to screen extracellular proteins in their genomes.

No.	Strains Name	AA	Gene ID	MW (Da)	SP	MW-SP	Conserved	Chr/Plasmid
1	*Bacillus cereus* Q1	174	221642232	18,540	2247.76	16,294.24	NlpC/P60	plasmid
2	*Bacillus thuringiensis* serovar pulsiensis BGSC	174	228918559	18,640	2419.97	16,220.03	NlpC/P60	Chr
3	*Ruminococcus obeum* ATCC 29174	334	153810514	36,687	3464.35	33,222.65	NlpC/P60	Master WGS * (no chromosome)
	*Clostridium bolteae* ATCC BAA-613	334	160936317	36,720	3482.38	33,237.62	NlpC/P60 famly; pfam 00877	Master WGS (no chromosome)
4	*Clostridium difficile* 630	334	126700956	36,623	3464.35	33,158.65	NlpC/P60 famly; pfam 00877	Chr
5	*Mollicutes bacterium* D7	334	237733575	36,641	3464.35	33,176.65	NlpC/P60 famly; pfam 00877	Master WGS (no chromosome)
6	*Lacnospiraceae bacterium* 3_1_57FAA_CT1	334	336429196	36,642	3464.35	33,177.65	NlpC/P60 famly; pfam 00877	Master WGS (no chromosome)
7	*Lactobacillus vini* DSM 20605	334	406838964	35,868	2990.79	32,877.21	NlpC/P60 famly; pfam 00877	No data
8	*Bacillus cereus* R309803	174	229162067	18,508	2233.74	16,284.26	NlpC/P60 famly; pfam 00877	chr
9	*Bacillus cereus* AH187	174	190015595	18,604	2281.78	16,322.22	NlpC/P60 famly; pfam 00877	plasmid
10	*Bacillus cereus* Rock1-15	174	229113559	18,692	2497.99	16,194.01	NlpC/P60 famly; pfam 00877	chr
11	*Bacillus cereus* BAG4X12-1	174	401113141	18,881.37	2526.04	16,355.33	NlpC/P60 famly; pfam 00877	Master WGS (no chromosome)
12	*Bacillus cereus* VD107	174	401239705	18,639.95	2467.96	16,171.99	NlpC/P60 famly; pfam 00877	Master WGS (no chromosome)
13	*Bacillus cereus* AH603	174	229065268	18,535	2437.94	16,097.06	NlpC/P60 famly; pfam 00877	chr
14	*Bacillus cereus* HuA2-1	174	402445052	18,678.13	2437.94	16,240.16	NlpC/P60 famly; pfam 00877	Master WGS (no chromosome)
15	*Bacillus cereus* VD142	181	401073234	19,521.11	2349.81	17,171.3	NlpC/P60 famly; pfam 00877	Master WGS (no chromosome)
16	*Bacillus cereus* MSX-D12	348	401201280		No			

AA = no of amino acids; MW = molecular weight; SP = signal peptide; chr = chromosome; * WGS = *whole genome* shotgun.

## Data Availability

Data are contained within the article.

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
