# Peer review of "Molecular Cloning, In Silico Analysis, and Characterization of a Novel Cellulose Microfibril Swelling Gene Isolated from Bacillus sp. Strain AY8"

_microorganisms, 2023, doi:10.3390/microorganisms11122857_

Round 1

Reviewer 1 Report

Comments and Suggestions for Authors

The study employs various computational and biochemical techniques, such as molecular docking, molecular dynamics simulations, and crystallinity measurements, to elucidate the mechanism of action and substrate specificity of the cms enzyme. The findings of this study are significant as they report the first bacterial cellulose microfibril swellase enzyme, which could have potential applications in the biofuel industry.

Suggestions for improvement:

The manuscript could benefit from a more detailed comparison of the cms enzyme with other known cellulose-degrading enzymes, highlighting the unique features and advantages of the cms enzyme.

It would be interesting to include additional experimental data on the effect of the cms enzyme on different types of cellulose substrates to further demonstrate its potential in biofuel production.

The discussion section could be expanded to include a more comprehensive overview of the potential applications and future prospects of the cms enzyme in biotechnology and bioengineering fields.

Overall, this manuscript reports a novel and significant discovery of a bacterial cellulose microfibril swellase enzyme, which has important implications for biofuel production and cellulose degradation. The study is well-designed, and the results are thoroughly analyzed and discussed.

Comments on the Quality of English Language

The English language used in this manuscript is generally of good quality, with clear and concise language used throughout.

Reviewer 2 Report

Comments and Suggestions for Authors

The article microorganisms-2671511 is devoted to the interesting topic of isolating and describing a new gene for swelling of cellulose microfibrils. Unlike previously published works, the authors chose Bacillus sp. as the object of research. strain AY8 was isolated from the pulp sludge of the paper mill (Moorim 131 Paper, JinjuCity, Korea). Based on the location of the bacterium, it can be assumed that some strains must have enzymatic activity aimed at hydrolyzing cellulose. The choice seems reasonable.

The positive aspects of the work include a detailed presentation of the results, for example, in lines 283-285, the authors indicate that when developing primers they encountered a discrepancy between theory and practice and successfully solved the problem of creating primers. Similarly, in lines 324-326, the actual molecular weight of the enzyme determined is different from the estimated molecular weight. Therefore, the authors understand that further theoretical considerations will also diverge from practical results.

The abstract, the main content of the article and conclusions cannot be considered proven and scientifically substantiated, which is what the authors themselves understand and what the authors write about in the conclusion. Currently, the presented article is scientific abstractionism, perhaps popular scientific reasoning. While publication in the journal Microorganisms involves the publication of results and discussion of experiments, and the experiments must be reproducible. Therefore, I suggest that the authors rewrite the article, remove sections 2.8-2.11 and 3.4 and expand section 3.5.

In addition, there are the following notes:

1. Abstract and section 3.5. A decrease in the crystallinity index by 3% does not go beyond the measurement error of the crystallinity index; it is not significant and cannot be interpreted as an achievement.

2. In Section 2.3, briefly describe what enzymatic activities Bacillus sp. has. strain AY8, what is currently installed?

3. Please use the French, 2014 method rather than the Segal, 1959 method to interpret the results of X-ray diffraction analysis.

French AD (2014) Idealized powder diffraction patterns for cellulose polymorphs. Cellulose 21(2): 885–896. https://doi.org/10.1007/s10570-013-0030-4

French AD (2020) Increment in evolution of cellulose crystallinity analysis. Cellulose, 27(10): 5445–5448. https://doi.org/10.1007/s10570-020-03172-z

4. In the supplementary materials to the article, please provide the results of X-ray diffraction analysis and infrared spectroscopy for all three materials used: filter paper, defatted medical grade cotton pulp and Avicel. And in section 3.5, present a detailed discussion of your results.

5. Neither X-ray diffraction analysis (an arbitrage method for determining the structure of cellulose) nor infrared spectroscopy can prove the destruction of hydrogen bonds in cellulose. How do the authors plan to further prove the action of the enzyme swellase described by them?

Reviewer 3 Report

Comments and Suggestions for Authors

The current manuscript deals with a topic of high interest, namely swellase enzyme. Even if it includes preliminary results that will desserves to be assessed by in vivo studies, its content could be reported. However, the current manuscript will require strong rewritings before acceptance: the English should be fully corrected and the manuscript will benefit from being concisely rewritten. Some figures will need to be moved to SI.

Comments on the Quality of English Language

The level of English is rather poor: please check it carefully.

Author Response

Reviewer 3

The current manuscript deals with a topic of high interest, namely swellase enzyme. Even if it includes preliminary results that will desserves to be assessed by in vivo studies, its content could be reported. However, the current manuscript will require strong rewritings before acceptance: the English should be fully corrected and the manuscript will benefit from being concisely rewritten. Some figures will need to be moved to SI.

Author Response: Thank you for pointing this out. I am sincerely grateful for the time you took to review our manuscript. Your thoughtful assessment and helpful feedback have significantly improved our work. The manuscript has been checked thoroughly by a native English speaker. 

Round 2

Reviewer 2 Report

Comments and Suggestions for Authors

The authors have fundamentally revised the content of the article. Essentially this is a new article with a new concept.

I have new questions:

1) In section 2.3, the authors indicate that Bacillus sp. strain AY8 was cultivated on a medium that decomposes filter paper. That is, the filter paper was the only source of carbon. At the same time, lines 146-147 indicate that cellulases and cellulolytic enzymes are excluded from the producer. This is a logical collapse.

2) The authors study the structure of alpha cellulose in Figures 8.1 and 8.2, but its description is missing in section 2.1. What is alpha cellulose and where is it produced?

3) In Figures 8.1 and 8.2, the black and red lines represent essentially parallel experiments. They can be obtained, for example, by varying the sample for analysis. They do not prove in any way the breaking of hydrogen bonds between sheets of cellulose. Thus, the authors' main claim does not appear to be substantiated.

4) The authors discuss polymer chemistry, so it is not clear why this article is classified in the journal Microorganisms and not in the journal Polymers?

Reviewer 3 Report

Comments and Suggestions for Authors

The current manuscript has been largely improved and desserves now to be reported.

Author Response

We expressed our sincere gratitude for taking the time to review the second version of our manuscript. Your constructive feedback played a crucial role in enhancing the quality of our work, and we truly appreciate your valuable insights. We are delighted to hear that you believe the manuscript now deserves to be reported.